# Bayesian Tabular Few-shot Learning with Causal Information

**Ole Ossen** [1]  **Jake Robertson** [1 2 3]  **Arik Reuter** [4 5]  **Magnus Bühler** [1]  **Lennart Purucker** [1 3]  **Frank Hutter** [1 2 3]

## Abstract

Recently, the task of tabular few-shot prediction has received considerable attention, with most existing methods incorporating contextual information using language models. In contrast, we present a purely Bayesian method for tabular few-shot prediction based on providing a probability distribution over possible causal structures as an additional model input. The model then performs Bayesian inference, weighing different data generation hypotheses by considering both their likelihood of generating the provided data and their compatibility with the provided distribution over causal structures. In our evaluations, we find that our method significantly increases model performance even on real world data when suitable causal information is provided.

## 1. Introduction

We consider the task of tabular few-shot learning, by which we mean the task of performing tabular prediction given fewer than 50 context samples. The difficulties of tabular prediction — for example, lack of a structural prior and heterogeneous features — are only exacerbated in this setting. Still, much work has been done on tabular few-shot learning in recent years.

One line of research assumes a semi-supervised setting, using additional unlabeled samples to create more robust embeddings (Yoon et al., 2020; Ucar et al., 2021; Bahri et al., 2022; Nam et al., 2023; Lee et al., 2025; Liu et al., 2024). Other approaches incorporate large language models (LLMs), the basic idea being to serialize tabular data into a textual input with which LLMs can be finetuned and prompted (Dinh et al., 2022). See Fang et al. (2024) for

[1]University of Freiburg, Freiburg im Breisgau, Germany [2]ELLIS Institute Tübingen, Tübingen, Germany [3]Prior Labs, Freiburg im Breisgau, Germany [4]Max Planck Institute for Intelligent Systems, Tübingen, Germany [5]University of Cambridge, Cambridge, United Kingdom . Correspondence to: Ole Ossen <ole.ossen@gmail.com>.

Proceedings of the $2^{nd}$ *ICML Workshop on Foundation Models for Structured Data*, Seoul, South Korea. 2026. Copyright 2026 by the author(s).

an overview of such tabular language models. As a rule, LLM-based tabular prediction is useful in the zero-shot and very-few-shot setting, but is of limited use with even slightly increased context size and suffers from typical drawbacks of LLMs like lack of reproducibility; see Gorla & Puduppully (2026) for a recent study that calls into question the generalization ability of LLM-based tabular models. Moreover, LLM-based tabular models are typically made for classification rather than regression tasks.

In recent years, the success of prior-data fitted networks (PFNs) has heralded a paradigm shift for tabular data. PFNs utilize *amortized Bayesian inference*: Models are pretrained on a large number of tabular prediction tasks and at inference time perform prediction through in-context learning (ICL). The resulting tabular foundation models (TFMs) like TabPFN (Hollmann et al., 2023; 2025; Grinsztajn et al., 2026), TabICL (Qu et al., 2025; 2026), and TabDPT (Ma et al., 2025) have achieved state-of-the-art performance in tabular prediction (Erickson et al., 2025). A recent attempt to unify the TFM paradigm and the ability of LLMs to process contextual information is the ContextTab model (Spinaci et al., 2025).

In this paper, we follow a completely different and purely Bayesian approach for improving the few-shot capabilities of TFMs: Our models take as an additional input a probability distribution over causal structures. A causal structure underlying a dataset $D$ is a directed acyclic graph (DAG) with one node for each feature of $D$ and one node for the target variable. Consider, for example, the following causal structure:

$$\text{(1)}$$

A lot of structural information is encoded in (1). To give just one example, conditional on "It rains", the variable "Wet road" is independent of the remaining two variables (Pearl, 2009, Section 1.2). This fact might be inferred by a TFM given sufficiently many training samples, but not necessarily in a few-shot setting. Thus providing (1) as an additional input we explicitly instruct the model to consider not only the compatibility of a given hypothesis with the training data, but also with the provided causal structure. The recent work of Reuter et al. (2026) also explores the effect of con-

ditioning tabular foundation models on causal information. The main differences to our work are as follows:

- While Reuter Reuter et al. (2026) are concerned with predicting interventional distributions, we focus on the equally relevant observational setting.
- We focus on the case of few-shot prediction and find that this is precisely the setting where conditioning on causal information has a large effect (cf. Section 4.1).
- While causal information is specified in Reuter et al. (2026) by means of a binary or ternary matrix, we investigate a more general scheme, where causal information is specified by a probabilistic matrix, which can make predictions more robust to graph misspecification (cf. Section 4.2).
- We include evaluation on real world data using causal discovery algorithms to provide an estimate of the causal graph and show that our method yields substantial gains.

We concentrate on the task of tabular regression (the prediction task less by existing methods). Overall, **our contributions** are: We describe a Bayesian framework for conditioning PFNs on causal information (Section 2), and implement it using several variants of a transformer-based architecture (Section 3). We find in synthetic case studies that our method works as expected. Furthermore, we observe that when provided a strong estimate of the causal graph, it significantly increases performance of tabular few-shot regression even on real-world data (Section 4).

## 2. Bayesian setup

### 2.1. Causal graph information

Consider data $x \in \mathbb{R}^{N \times F}$, $y \in \mathbb{R}^N$, where $N$ denotes the number of samples and $F$ the number of features. By a piece of *causal graph information* underlying $(x, y)$ we mean a probability distribution over DAGs on the vertex set $V = \{x_1, \ldots, x_F, y\}$. We parametrize causal graph information by *probabilistic adjacency matrices*. A probabilistic adjacency matrix is a matrix

$$\gamma \in [0,1]^{V \times V} \quad \text{such that} \quad \gamma + \gamma^T \in [0,1]^{V \times V}. \quad (2)$$

The probability distribution associated with $\gamma$ is defined as follows: For each pair of nodes $(i, j)$, add an edge $i \rightarrow j$ with probability $\gamma_{ij}$; add an edge $j \rightarrow i$ with probability $\gamma_{ji}$; do not add an edge with probability $1 - \gamma_{ij} - \gamma_{ji}$, this value being non-negative because of the assumption in (2). If the resulting directed graph is not acyclic, we must discard it and restart the sampling process. We explore three different ways of creating graph information $\gamma$:

1) **Binary graph information**. In this setting, $\gamma \in \{0, 1\}^{V \times V}$. Thus $\gamma$ determines a unique graph.
2) **Beta type graph information**. The entries of $\gamma$ are

distributed according to a beta distribution. Thus for most pairs of nodes $(i, j)$, the probability of an edge between $i$ and $j$ is either close to 1 or close to 0.

3) **Uniform type graph information**. The entries of $\gamma$ are distributed according to a uniform distribution. This type of graph information entails the greatest uncertainty about the graphs sampled from it.

### 2.2. SCM priors

We consider data generating processes based on structural causal models (SCMs). An SCM is determined by an underlying DAG $G$ as well as a structural equation

$$x_i = f_i(pa_i, \varepsilon_i)$$

for each node $i$ of $G$. Here $pa_i$ denotes the concatenation of all parent features of node $i$, and $\varepsilon_i$ a noise term. Overall, we sample datasets in a four-stage process (cf. Figure 1):

1) Sample a piece of graph information $\gamma$
2) Sample a DAG $G$ from $\gamma$
3) Sample an SCM $\sigma$ on this underlying DAG $G$
4) Sample a dataset $D$ from $\sigma$

Taken together, these steps define a probability distribution $p(D)$ over datasets, which decomposes naturally as

$$p(D) = \int_\sigma \int_\gamma p(D \mid \sigma) p(\sigma \mid \gamma) p(\gamma).$$

We refer to Appendix A for details of the procedure.

### 2.3. Posterior distributions

Given a context dataset and a query sample $D, (x, y) \sim p(D)$, we want to compute their associated posterior predictive distribution (PPD)

$$p(y \mid x, D) = \int_\sigma p(y \mid x, \sigma) p(\sigma \mid x, D). \quad (3)$$

The strategy of the present work is to modify (3) by conditioning additionally on a piece of graph information $\gamma$, yielding the *graph-conditioned PPD*

$$p(y \mid x, D, \gamma) = \int_\sigma p(y \mid x, \sigma) p(\sigma \mid x, D, \gamma). \quad (4)$$

Under (4), each possible SCM $\sigma$ is now weighed not only according to its compatibility with the data $D$, but also according to its compatibility with the graph information $\gamma$. As the size of the dataset $D$ increases, it will begin to overpower the prior, until the difference between (3) and (4) disappears:

**Insight 2.1.** *Under certain technical conditions, the distributions $p(y \mid x, D)$ and $p(y \mid x, D, \gamma)$ converge to the same distribution as the size of $D$ goes to infinity (see Appendix B.1 for a precise statement).*

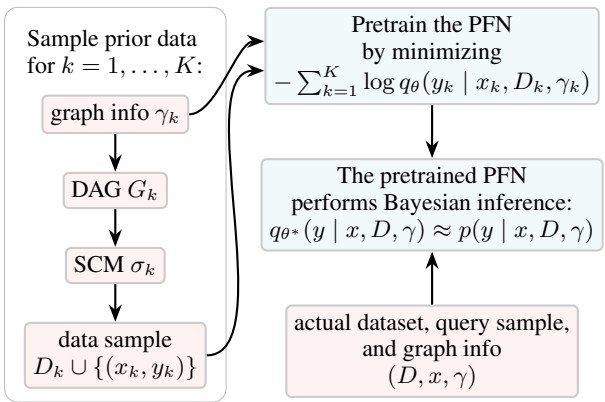

Figure 1. Overview of the pretraining process, based on an analogous illustration in (Müller et al., 2022).

This insight explains our focus on few-shot learning. We verify experimentally in Section 4.1 that graph-conditioning has the largest effect when the context size is small, with diminishing effects as the context size grows.

## 3. Methodology

### 3.1. Pretraining procedure

To pretrain a model approximating the graph-conditioned PPD (4), we employ variations of the TabPFNv2 architecture (Hollmann et al., 2025) that also take graph information $\gamma$ as an input. The pretraining scheme is visualized in Figure 1. It is analogous to the pretraining procedure established in Müller et al. (2022): After sampling a piece of graph information $\gamma$, a dataset $D$, and a query $(x, y)$, the transformer model $q_\theta$ produces a negative log-likelihood (NLL) $-\log q_\theta(y \mid x, D, \gamma)$. The parameter set $\theta$ is then updated to minimize this NLL via backpropagation. A more detailed description of the pretraining algorithm and hyperparameter choices may be found in Appendix D.

The validity of this approach for approximating the graph-conditioned PPD (4) is established by the following graph-conditioned analog of (Müller et al., 2022, Insight 1), whose proof may be found in Appendix B.2:

**Insight 3.1.** *In expectation, the NLL of the model output agrees with the cross-entropy between the model output distribution and the graph-conditioned PPD:*

$$\mathbb{E}_{D,x,y,\gamma}\big[-\log q_\theta(y \mid x, D, \gamma)\big]$$
$$= \mathbb{E}_{D,x,\gamma}\Big[H\big(p(\cdot \mid x, D, \gamma), q_\theta(\cdot \mid x, D, \gamma)\big)\Big]$$

### 3.2. Model architecture

Recall that the backbone of the TabPFNv2 architecture (Hollmann et al., 2025) consists of alternating *feature-wise attention* and *sample-wise attention* layers with interspersed

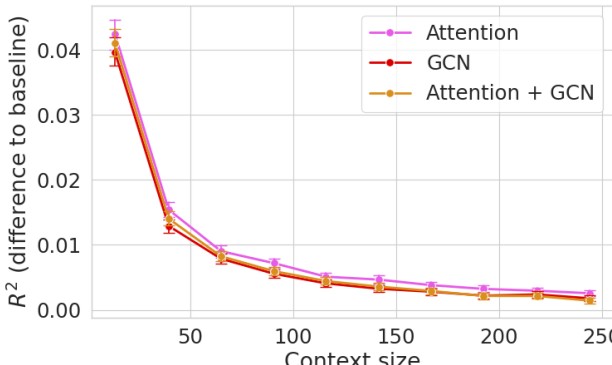

Figure 2. Model performance as a function of context size of three graph-conditioning architectures over non-graph-conditioned baseline. Mean and bootstrapped 95% CIs are shown. The attention-based model performs the best by a small margin. All models substantially outperform the baseline, but this difference becomes smaller and smaller with increasing context size.

layer norm layers (Ba et al., 2016). We explore two distinct methods of modifying the architecture to accept a piece of graph information $\gamma$ as an additional input:

1) **Causally restricted attention.** We penalize the attention weights of all feature-wise attention layers according to $\gamma$, with the penalty the greater the lower the probability assigned to a given edge by $\gamma$.
2) **GCN encoding and modulation using adaptive layer norm.** We globally encode the graph information $\gamma$ using a graph-convolutional network (GCN) and use the output to modulate the layer norm layers (Kipf & Welling, 2017; Peebles & Xie, 2023).
3) The two methods can also be used in conjunction.

More details on the model architectures can be found in Appendix C.

## 4. Results

We train a total of nine graph-conditioning models, one for each combination of graph-conditioning method (attention, GCN, attention plus GCN) and graph information type (binary, beta, uniform). Additionally, we train a non-graph-conditioned baseline model. Except for the graph-conditioning mechanism, all models have the same architecture, and all models see the same number of datasets from the same distribution during pretraining.

### 4.1. Evaluation on synthetic data

We evaluate our models that were trained on beta type graph information in-distribution on prior data. Figure 2 reports the results as a function of the context size. We see that graph-conditioning increases model performance substantially for all architectures. However, the difference between

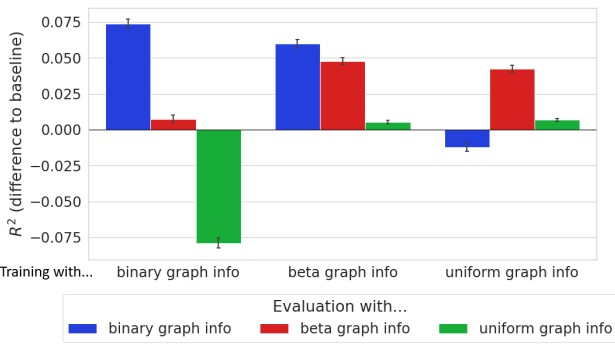

*Figure 3.* Evaluation of models with graph information of a different type from the one they were trained on. Context size is sampled uniformly from $\{1, \ldots, 20\}$. The three colors indicate the three types of graph information used for evaluation, the three sets of bars indicate three different models. The error bars show bootstrapped 95% CIs. We see that the model trained with the strongest type (binary) graph information achieves the best overall score. However, the models trained with weaker graph information (beta or uniform) are more robust when evaluated on the "wrong" type of graph information.

graph-conditioned models and baseline decreases with sample size. This is expected from a Bayesian standpoint (cf. Insight 2.1): With little context available, the model relies on the prior for its predictions and is helped much by graph information, while with more context available, the data overpowers the prior, and graph information is no longer helpful. A more detailed evaluation, involving models trained on different uncertainty levels, and showing performance as a function of various other dataset characteristics, may be found in Appendix E.1.

### 4.2. Different uncertainty levels

To investigate the effect of using different types of graph information, we take three of our models with identical architecture (attention-based graph-conditioning), one trained on each type of graph information. We then evaluate each model on each type of graph information.

The results are summarized in Figure 3. We see that the overall best scores are achieved by the model trained and evaluated on binary graph information. This is expected, since binary graph information determines a DAG completely. However, this model's performance is bad when a different type of graph information is provided. The model trained with beta type graph information is more robust; it achieves good scores with both binary and beta type graph information. Finally, the model trained with uniform type graph information underperforms the baseline when provided with binary graph information, but performs well with other types. See Appendix E.2 for an additional evaluation showing that models trained with binary graph information are more sensitive to misspecification of graph information.

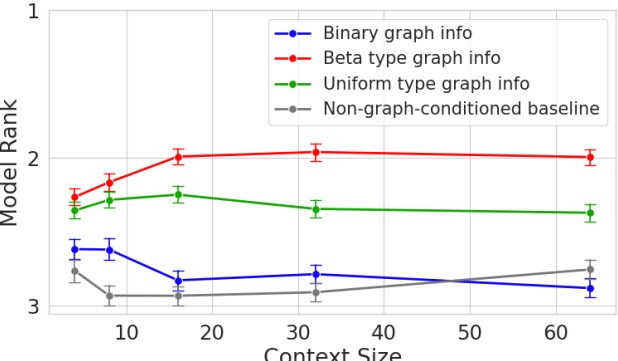

*Figure 4.* Evaluation of our models across eleven TabArena regression datasets. For each dataset and context size, 100 datasets are subsampled for evaluation. We report average model rank (to achieve a meaningful aggregation of the results for the different datasets), and bootstrapped 95% CIs. Our graph-conditioning models outperform the non-graph-conditioning baseline, with the model trained on beta type graph information performing best.

### 4.3. Evaluation on real world data

To test the effectiveness of our methods on real world data, we apply our models to eleven TabArena regression datasets (Erickson et al., 2025). Since there is no ground-truth causal information available, we run an ensemble of causal discovery methods on a 500 row subsample of each dataset to obtain a fixed causal structure for each dataset. Since this uses a large subsample, the obtained causal information is in a sense "oracular"; in a real deployment setting, it would have to be provided by a domain expert or another source.

Results for three of our models with identical architecture (attention-based graph-conditioning) are summarized in Figure 4. We see that the graph-conditioning models consistently outperform the non-graph-conditioning baseline, with the model trained on beta type graph information performing best. We refer to Appendix E.3 for details on the datasets used, the causal discovery methods, and a more elaborate evaluation.

## 5. Discussion

We present a Bayesian way of conditioning PFNs on causal information, orthogonal to existing few-shot methods. Our evaluations empirically validate the theory and show that our method can be useful for real world tabular data. The strategy of conditioning the model on graph information generalizes to any foundation model whose prior consists of a sampling process with several stages, suggesting many interesting directions for future research: For example, an additional input to a TFM might be the types of functional mechanisms a dataset is known to follow; an additional input to a time series model might be a set of known relations between variates of a multivariate time series.

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

# A. SCM prior details

As mentioned in Section 2.2, the sampling of datasets from our SCM prior proceeds in four stages: 1) sampling graph information, 2) sampling a DAG, 3) sampling an SCM, and 4) sampling a dataset. Below, we give details on all four stages.

## A.1. Sampling graph information

First we sample the number of nodes uniformly from $\{3, \ldots, 25\}$ and a mean edge probability $\mu_e$ log-uniformly from the interval $[0.05, 0.5]$. To obtain a piece of graph information from these, we need two auxiliary distributions:

- an *edge probability distribution $E$* with support contained in the interval $[0, 1]$ and mean $\mu_e$
- a *direction probability distribution $D$* with support contained in the interval $[0, 1]$ and mean $0.5$

For each pair of nodes $i < j$, we then sample $e_{ij} \sim E$ and $d_{ij} \sim D$ and set

$$\gamma_{ij} = d_{ij}e_{ij}, \qquad \gamma_{ji} = (1 - d_{ij})e_{ij}.$$

The distribution $E$ assigns a probability to the pair of nodes $(i, j)$; the distribution $D$ assigns a probability to the two possible directions $i \rightarrow j$ and $j \rightarrow i$. We set $\gamma_{ii} = 0$ for all $i$, to avoid sampling graphs with loops.

The specific choice of $D$ and $E$ depends on the specific mode of creating graph information ("binary", "beta", "uniform"), as laid out in Table 1. Examples of resulting matrices are shown in Figure 5.

*Table 1.* Overview of the distributions defining graph information $\gamma$ for three different graph information types.

|  | edge distribution $E$ | direction distribution $D$ |
|---|---|---|
| binary | $\mathrm{Bernoulli}(\mu_e)$ | $\mathrm{Bernoulli}(0.5)$ |
| beta | $\mathrm{Beta}(\alpha, \beta)$, where $\alpha = \frac{\mu_e}{2(1-\mu_e)}, \beta = 0.5$ | $\mathrm{Uniform}([0, 1])$ |
| uniform | $\mathrm{Uniform}([0, 2\mu_e])$ | $\mathrm{Uniform}([0, 1])$ |

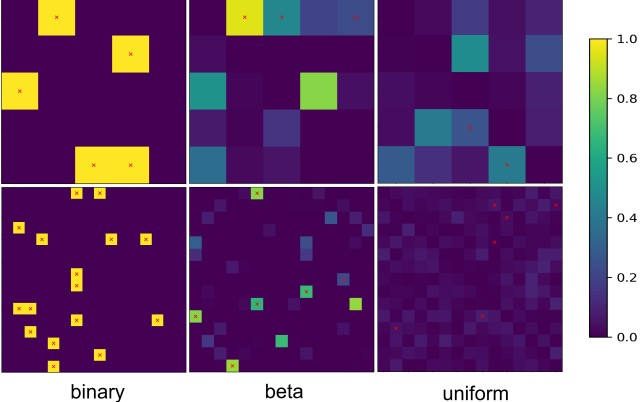

*Figure 5.* Examplary samples of the probabilistic adjacency matrices $\gamma$ for the different uncertainty levels of graph information. Brighter squares denote higher probabilities. The small red crosses indicate edges that were actually sampled.

## A.2. Sampling a DAG

As explained in Section 2.1, we proceed as follows given a piece of graph information $\gamma$. For each pair of nodes $(i, j)$, we sample $t$ uniformly at random from the interval $[0, 1]$. If $t < \gamma_{ij}$, we add an edge $i \rightarrow j$. If $\gamma_{ij} \leq t < \gamma_{ij} + \gamma_{ji}$, we add an edge $j \rightarrow i$. Otherwise we add no edge between $i$ and $j$.

If the resulting graph is not acyclic, we discard it and restart the entire procedure by sampling a new piece of graph information.

### A.3. Sampling an SCM

Given a DAG, we sample for each node in the DAG a noise standard deviation $\sigma_i$, an activation function $g_i$, as well as weights and biases. The noise standard deviation is sampled from a shifted exponential distribution with density function

$$x \mapsto \begin{cases} \frac{1}{\mu} \exp\left(\frac{s-x}{\mu}\right), & x \geq s, \\ 0, & \text{otherwise.} \end{cases}$$

For root nodes of the DAG (i.e. nodes with no parents), the mean and shift parameters are $\mu = 1$, $s = 0.2$; for non-root notes, they are $\mu = 0.1$, $s = 0.1$. The activation function $g_i$ is chosen uniformly at random from the list of activation functions used in TabICL (Qu et al., 2025, Section C.1). Moreover, we replace $g_i$ with the function $x \mapsto -g_i(x)$ with probability 0.5. Finally, we choose for each node $i$ and each parent $j$ of $i$ a bias $b_{ij}$ and weights $w_{ij}$ uniformly from $[-1, 1]$. The mechanism associated with node $i$ is then given by

$$x_i = f_i(pa_i, \varepsilon_i) = \text{arsinh}\left(g_i\left(\sum_j w_{ij}x_j + b_{ij}\right)\right) + \varepsilon_i,$$

where $\varepsilon_i \sim \mathcal{N}(0, \sigma_i)$. The role of the function $\text{arsinh}$ is to avoid datasets with extreme outliers by pulling all values back towards 0.

### A.4. Sampling a dataset

Having sampled an SCM, we proceed as follows to sample a dataset from it. First, the number of context samples is chosen uniformly at random from $\{1, \ldots, 256\}$. The number of query samples is fixed to 100. For each of these samples, data is obtained from the SCM via ancestral sampling. The result is a data table with one column for each node in the DAG underlying the SCM and between 101 and 356 rows.

Next, each column of the data table is standardized via (Bessel-corrected) z-normalization.

Finally, each column is made categorical with a probability of 0.15. If so, a number of values $c$ is chosen uniformly from $\{2, \ldots, 5\}$, and the column is bucketized into $c$ buckets: We choose a partition

$$\mathbb{R} = (-\infty, b_1] \cup (b_1, b_2] \cup \ldots \cup (b_c, \infty)$$

with each subinterval containing the same amount of elements, and then each subinterval is assigned one of the integer labels $0, 1, \ldots, c-1$.

One column of the resulting data table is chosen as the target $y$; the other columns are the features. Always, $y$ is chosen to not be a categorical column (which would not make sense in a regression task) and is chosen to not be isolated in the underlying DAG (that is, the node corresponding to the target column has children or parents).

## B. Proofs

### B.1. Proof of Insight 2.1

To formalize Insight 2.1, we use a Bayesian nonparametrics setup similar to Nagler (2023). We suppose that all SCMs we consider lie in an infinite-dimensional parameter space with probability measure $\Pi$. The support of the prior distribution over SCMs is a subspace, which we denote by $\mathcal{P}$.

Consider an infinite series of independent and identically distributed random variables $X, X_1, X_2, \ldots$, all with distribution following a fixed SCM $\sigma_0$. A joint sample of this sequence yields a sequence of feature vector and target pairs $(x, y), (x_1, y_1), (x_2, y_2), \ldots$. By truncating this sequence after $N$ steps, we get a query sample $(x, y)$ and an increasing sequence of datasets $D_N = \{(x_i, y_i) \mid i = 1, \ldots, N\}$.

Nagler provides a sufficient condition under which the PPD with respect to $D_N$ concentrates near the distribution given by a single SCM as $N \to \infty$ (Nagler, 2023, Appendix A.2):

(A1) There is a unique $\sigma^* \in \mathcal{P}$ with $\sigma^* = \arg\min_\sigma D_{KL}(\sigma_0 \mid \sigma)$ with $D_{KL}(\sigma_0 \mid \sigma^*) < \infty$.

(A2) For every $\alpha \in (0, 1/2)$, there is a covering

$$\mathcal{P} \subseteq \bigcup_{j \geq 1} B_j, \qquad \sup_{\sigma, \sigma' \in B_j} H(\sigma, \sigma') \leq 4(\alpha^2/2)^{1/\alpha}, \qquad \sum_{j \geq 1} \Pi(B_j)^\alpha < \infty,$$

where $H(\sigma, \sigma')$ denotes the Hellinger distance between $\sigma$ and $\sigma'$.

If (A1) and (A2) hold, then for almost every joint sample from $X, X_1, X_2, \ldots$ we have

$$p(y \mid x, D_N) \stackrel{N \to \infty}{\longrightarrow} p(y \mid x, \sigma^*) \tag{5}$$

(Nagler, 2023, Theorem 3.1). Now fix a piece of graph information $\gamma$ in the sense of Section 2.1. Denote by $\mathcal{P}_\gamma$ the support of the prior distribution over SCMs graph-conditioned on $\gamma$. Then, assuming the analogs of (A1) and (A2) with $\mathcal{P}$ replaced by $\mathcal{P}_\gamma$ hold, we get $p(y \mid x, D_N, \gamma) \stackrel{N \to \infty}{\longrightarrow} p(y \mid x, \sigma^*)$. Together with (5), this gives the desired identity.

### B.2. Proof of Insight 3.1

The proof is completely analogous to the proof of (Müller et al., 2022, Insight 1). Note that the expectations in both sides of the equality in Insight 3.1 are with respect to the joint density over graph data $\gamma$ and datasets — as explained in detail in Appendix A, $\gamma$ is sampled first and then $D, x, y$ are sampled from an SCM based on $\gamma$. Writing the expectations as integrals, we have:

$$\begin{aligned}
\mathbb{E}_{D,x,y,\gamma}\big[ - \log q_\theta(y \mid x, D, \gamma)\big] &= - \int_{D,x,y,\gamma} p(D, x, y, \gamma) \log q_\theta(y \mid x, D, \gamma) \\
&= - \int_{D,x,\gamma} p(D, x, \gamma) \int_y p(y \mid x, D, \gamma) \log q_\theta(y \mid x, D, \gamma) \\
&= \int_{D,x,\gamma} p(D, x, \gamma) H\big(p(\cdot \mid x, D, \gamma), q_\theta(\cdot \mid x, D, \gamma)\big) \\
&= \mathbb{E}_{D,x,\gamma}\Big[ H\big(p(\cdot \mid x, D, \gamma), q_\theta(\cdot \mid x, D, \gamma)\big)\Big]
\end{aligned}$$

## C. Details on model architecture

### C.1. Overall model architecture

The overall structure of all our models is identical; they differ only in the way that the graph information is incorporated into the transformer encoder layers. The architecture is based on the TabPFNv2 architecture (Hollmann et al., 2025) as implemented in the TFM Playground repository (Pfefferle et al., 2025). The non-graph-conditioned baseline models and the purely attention-based models have 5.4M trainable parameters. The GCN-based models have 6.7M trainable parameters.

Let us write $B$ for the batch size, $N$ for the number of samples, and $F$ for the number of features. Then the input to our models consists of a feature tensor $x \in \mathbb{R}^{B \times N \times F}$, a target tensor $y \in \mathbb{R}^{B \times N}$, a positive integer $1 \leq n < N$ indicating the position of the train/test split, and a probabilistic adjacency matrix $\gamma \in \mathbb{R}^{(F+1) \times (F+1)}$ (the latter is ignored by the non-graph-conditioned baseline model). We embed both $x$ and $y$ using linear layers $\mathbb{R} \to \mathbb{R}^D$ and concatenate the result to obtain a tensor in $\mathbb{R}^{B \times N \times (F+1) \times D}$.

Next, we apply six transformer encoder layers; this is the core part of the model and the only part where the different graph conditioning methods differ. For the non-graph-conditioning baseline model, each transformer encoder layer consists of a *feature-wise* or "horizontal" attention layer, followed by a layer norm layer, followed by a *sample-wise* or "vertical" attention layer, followed by a second layer norm layer, an MLP, and a final layer norm layer. All attention layers use eight heads.

The encoded tensor is sliced to obtain a tensor of shape $\mathbb{R}^{B \times (N-n) \times D}$ and decoded by an MLP. The result is a logit tensor of shape $\mathbb{R}^{B \times (N-n) \times 2000}$. The 2000 classes correspond to 2000 evenly spaced buckets partitioning the interval $[-10, 10]$. In this way, the model output specifies a bar distribution (Müller et al., 2022, Sections 4 and E.2).

### C.2. Graph conditioning mechanisms

As mentioned in Section 3.2, we explore two distinct mechanisms to condition on graph information. They can also be used in conjunction, for a total of three different graph-conditioning architectures. Also taking into account the three uncertainty levels (binary, beta, uniform), we get nine different graph-conditioning models. See Table 2 for an overview.

For the **attention-based graph-conditioning**, the only thing changed in comparison to the baseline model is the feature-wise attention layers. We create two masks in log-space

$$M_1 = \log(\gamma + I_{F+1}), \quad M_2 = \log(\gamma^T + I_{F+1}) \quad \in \quad [-\infty, 0]^{(F+1)\times(F+1)}.$$

The eight attention heads are split into two groups of four. In the first group, $M_1$ is added to the attention weights of the attention mechanism; in the second group, $M_2$ is added. In the first group, attention of feature $i$ to feature $j$ is discouraged if the $(i, j)$-entry of $M_1$ is very negative. This happens precisely when $\gamma_{ij}$ is close to zero, that is, when an edge $i \rightarrow j$ in the DAG is judged unlikely by $\gamma$. Similarly, in the second group, attention of feature $i$ to feature $j$ is discouraged if an edge $j \rightarrow i$ is judged unlikely. We add the identity matrices $I_{F+1}$ so that every feature is allowed to attend to itself.

For the **GCN-based graph-conditioning**, we encode $\gamma$ using a graph-convolutional network (Kipf & Welling, 2017). We begin with computing normalized versions of $\gamma$ and its transpose,

$$A_1 = \deg(\gamma + I_{F+1})^{-1}(\gamma + I_{F+1}), \qquad A_2 = \deg(\gamma^T + I_{F+1})^{-1}(\gamma^T + I_{F+1}),$$

where $\deg(M)$ denotes the degree matrix of $M$ (the diagonal matrix whose entries are the row sums of $M$). A GCN layer then takes the form

$$z \mapsto f_1(A_1 z) + f_2(A_2 z),$$

where $f_1$, $f_2$ are linear layers. To encode $\gamma$, we initialize $z$ using a standard positional encoding (Vaswani et al., 2017) and then apply two GCN layers with interspersed layer norm layer and ReLU activation. The encoding of $\gamma$ is used as an input for modulating the layer norms in the transformer encoder described in the previous two sections: Instead of normal layer norm layers, we use learnable adaptive layer norm layers (Peebles & Xie, 2023) controlled by the encoding of $\gamma$.

The two methods for conditioning on graph information (attention and GCN) can also be used in conjunction: In this setting, we simply use both strategies discussed above: The attention weights of the feature-wise attention layers are penalized by $\gamma$ and the layer norm layers are modulated by the GCN encoding of $\gamma$.

*Table 2.* An overview of the ten models we trained. Each row corresponds to a different type of graph information, each column to a different graph-conditioning architecture. For the baseline model, there is no distinction with respect to graph information.

| attention binary | GCN binary | attention+GCN binary | |
|---|---|---|---|
| attention beta | GCN beta | attention+GCN beta | baseline |
| attention uniform | GCN uniform | attention+GCN uniform | |

# D. Pretraining details

All our models were trained on 2,000,000 batches consisting of four datasets each. We use a schedule-free AdamW optimizer (Defazio et al., 2024) with a learning rate of $10^{-4}$. Training one of our models in this way takes about 70 hours on a single RTX 2080 GPU. A pseudocode description of the pretraining procedure can be found in Algorithm 1.

# E. Additional evaluations

## E.1. Additional evaluations on synthetic data

The evaluation on synthetic data in Section 4.1 was for the sake of conciseness restricted to our models trained on beta type graph information. We now extend it to binary and uniform type graph information. In other words: While the evaluation in Section 4.1 was limited to the second row of Table 2, we now include the entire table. Moreover, we show model performance as a function not only of context size, but also of several other dataset characteristics: number of features, density of the causal graph, and average noise level of the SCM. The graph density is measured by the "edge probability", which is the parameter called $\mu_e$ in Section A.1. The noise level is measured by the mean of the standard deviations denoted $\sigma_i$ in Section A.3.

Results are shown in Figure 6. We observe the following:

- The effect of the graph-conditioning mechanism (attention, GCN, or attention + GCN) is small. Still, the purely attention-based architecture consistently performs best.

**Algorithm 1** The pretraining algorithm used for training our models. Except for our more elaborate sampling procedure involving graph information, it is standard (Müller et al., 2022, Algorithm 1).

---

**Input:** number of batches $K$ and batch size $B$; distributions $p(\gamma)$, $p(\sigma \mid G)$, $p(D \mid \sigma)$ that we can sample from; an untrained model $q_\theta$
**Output:** the trained model $q_\theta$
**for** $k = 1, 2, \ldots, K$ **do**
   Sample $\gamma \sim p(\gamma)$   `// Section A.1`
   Sample $G \sim p(G \mid \gamma)$   `// Section A.2`
   Sample $\sigma \sim p(\sigma \mid G)$   `// Section A.3`
   **for** $b = 1, 2, \ldots, B$ **do**
      Sample $D^{(b)} \cup \{(x_i^{(b)}, y_i^{(b)})\}_{i=1}^m \sim p(D \mid \sigma)$   `// Section A.4`
   **end for**
   Compute loss $\mathcal{L}(\theta) = \sum_{b=1}^B \sum_{i=1}^m - \log q_\theta(y_i^{(b)} \mid x_i^{(b)}, D^{(b)}, \gamma)$
   Update $\theta$ by gradient descent on $\mathcal{L}$
**end for**

---

- Performance as a function of dataset characteristics follows similar trends for all nine models. However, gains over the baseline are largest for the models trained with binary graph information, are somewhat smaller for models trained with beta type graph information, and are almost nonexistent for models trained with uniform type graph information. (Note however, that there is a consistent and statistically significant improvement even for models trained with uniform type graph information in the case of very small context size.)
- As was already remarked in Section 4.1, improvement over the baseline decreases with increasing context size. Improvement over the baseline also decreases with increasing graph density, but increases with the number of features, and is a concave function of the noise level.

### E.2. Additional evaluation on underspecified causal information

To further test how robust models trained on different types of graph information are to misspecified graph information, we evaluate our models in-distribution, but with a fraction of the entries of $\gamma$ replaced by the entries of the matrix

$$\frac{1}{3} \begin{pmatrix} 0 & 1 & \cdots & 1 \\ 1 & \ddots & \ddots & \vdots \\ \vdots & \ddots & \ddots & 1 \\ 1 & \cdots & 1 & 0 \end{pmatrix}. \tag{6}$$

This matrix expresses a complete lack of prior belief about the causal structure. For each pair of nodes $i < j$, we assign equal probability $1/3$ to there being an edge $i \to j$, there being an edge $j \to i$ and there being no edge between $i$ and $j$.

Results for three of our models with identical architecture (attention-based graph-conditioning) are summarized in Figure 7. They are similar to the results of Section 4.2. While the model trained with binary graph information performs the best when given the correct and unmodified causal information, its performance also degrades the most when given perturbed causal information. In contrast, the model trained with beta type graph information is more robust. The model trained with uniform type graph information always performs similar to (or slightly below) the baseline, even when the graph information is completely replaced with (6).

### E.3. Additional evaluations on real world data

For our evaluations on real world data both in Section E.3 and here, we use eleven regression datasets from the TabArena benchmark (Erickson et al., 2025): QSAR fish toxicity, concrete_compressive_strength, healthcare_insurance_expenses, airfoil_self_noise, Another-Dataset-on-used-Fiat-500, wine_quality, miami_housing, houses, Food_Delivery_Time, physio-chemical_protein, and diamonds. We exclude the remaining two regression datasets (QSAR-TID-11 and superconductivity), their large number of features making them unsuitable for models trained on our prior without adjustment.

For the evaluation in Section 4.3, we used an ensemble of four causal discovery methods as implemented by Zheng et al.

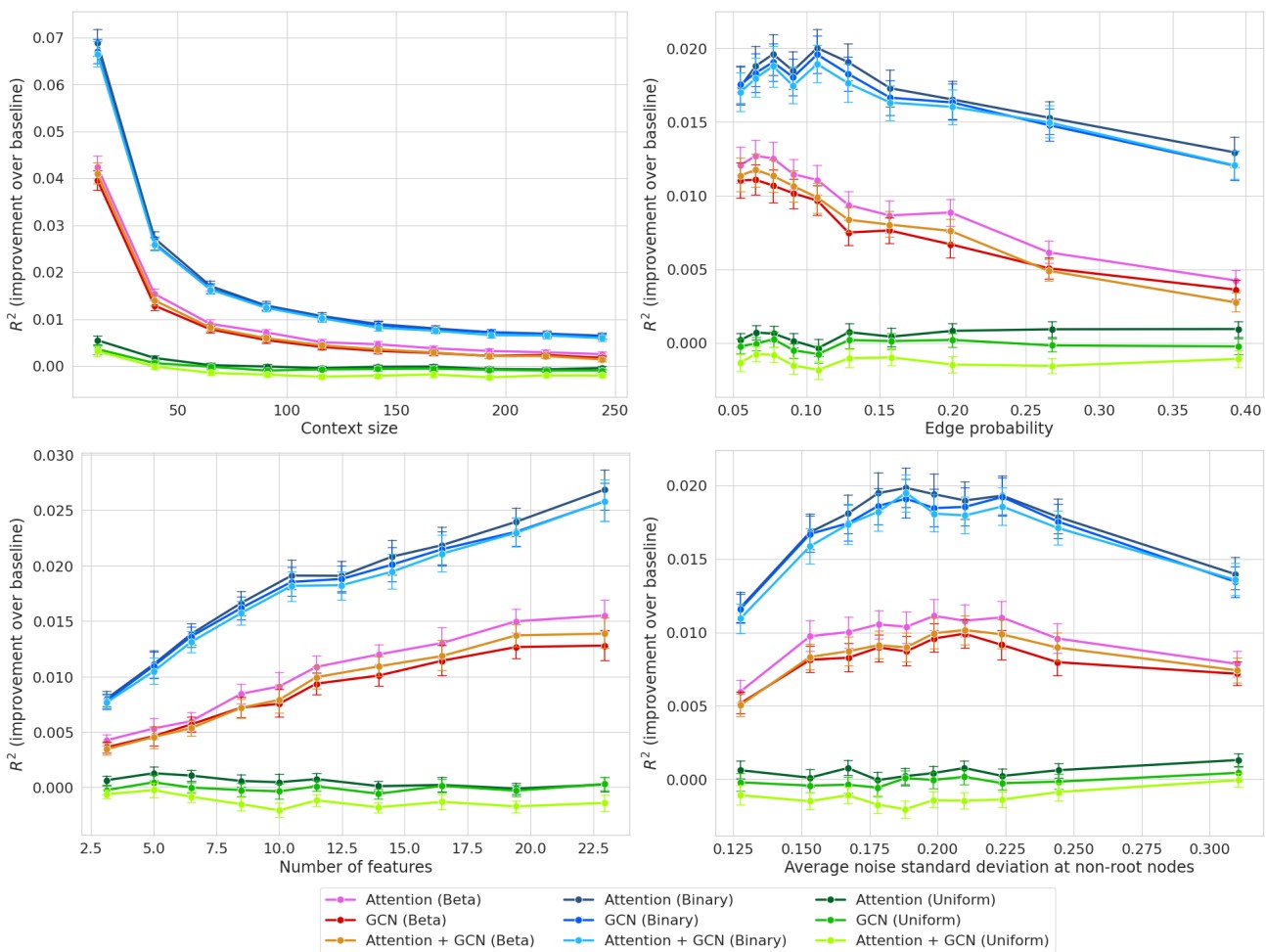

*Figure 6.* Model performance as a function of four different dataset characteristics: context size, edge probability in the causal graph, number of features, and the average standard deviation of the noise distributions at non-root nodes of the causal graph. Mean improvement over the baseline and bootstrapped 95% CIs are shown. The three blue lines correspond to the models trained with binary graph information, the red lines to models trained with beta type graph information, the green lines to models trained with uniform type graph information. We see that the difference to the baseline is greatest for the models trained on binary graph information, and that the difference is very small for the models trained on uniform type graph information. The difference to the baseline decreases with context size and edge probability, increases with number of features, and is a concave function of the noise level.

(2024):

- PC algorithm (Spirtes et al., 1993, Section 5.4.2)
- FCI algorithm (Spirtes et al., 1995)
- GES algorithm (Chickering, 2003)
- ICA-LiNGAM algorithm (Shimizu et al., 2006)

For each dataset, this ensemble is run on a subsample of 500 rows. This results in one fixed causal structure for each dataset, which was used for all evaluations. As mentioned in Section 4.3, this strategy is not straight-forwardly usable in a real deployment setting. However, we believe that it is a fair approximation of causal structure input by a domain expert. An alternative strategy would be to run the causal discovery ensemble on only the few-shot context samples. However, we found that this does not work better than simply providing the constant matrix (6) as an input. Results for three of our models (attention-based graph-conditioning) on each of the TabArena datasets are shown in Figure 8. We have also included the ContextTab model (Spinaci et al., 2025) in the evaluation. We observe the following:

- As already remarked in Section 4.3, our graph-conditioned models using causal information obtained by the causal discovery ensemble usually outperform the non-graph-conditioned baseline. The model trained on beta type graph

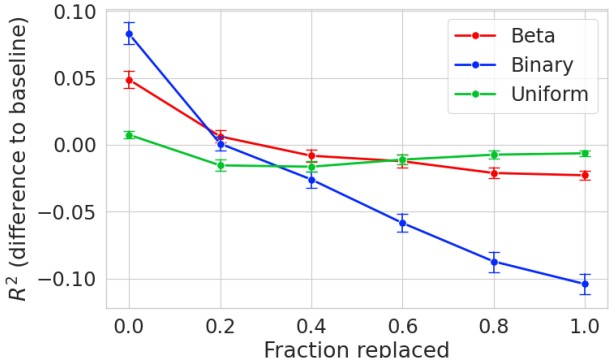

*Figure 7.* Model performance as a function of the fraction of entries of $\gamma$ replaced by the corresponding entry in the constant matrix (6), with context size sampled uniformly from $\{1, \ldots, 20\}$. We show mean improvement over the non-graph-conditioned baseline for three models with identical architecture (attention-based graph-conditioning) and bootstrapped 95% CIs. As expected, model performance degrades with the number of replaced entries. The model trained on binary graph information is the least robust. The model trained with beta type graph information is more robust, only dropping below the baseline by a small margin. The model trained on uniform type graph information never performs well, but also doesn't degrade further as more entries are replaced.

information performs best overall.
- The models trained with graph-conditioning, but provided the constant matrix (6) at inference time perform surprisingly well, often beating the non-graph-conditioned baseline. This is in contrast to evaluations on synthetic data (Section E.2), where providing the constant matrix (6) yields subpar results.
- The ContextTab model typically performs worse than our models for very small sample sizes. As context size increases, it overtakes them at some point (which we attribute to ContextTab's training on a much broader prior of real world data).

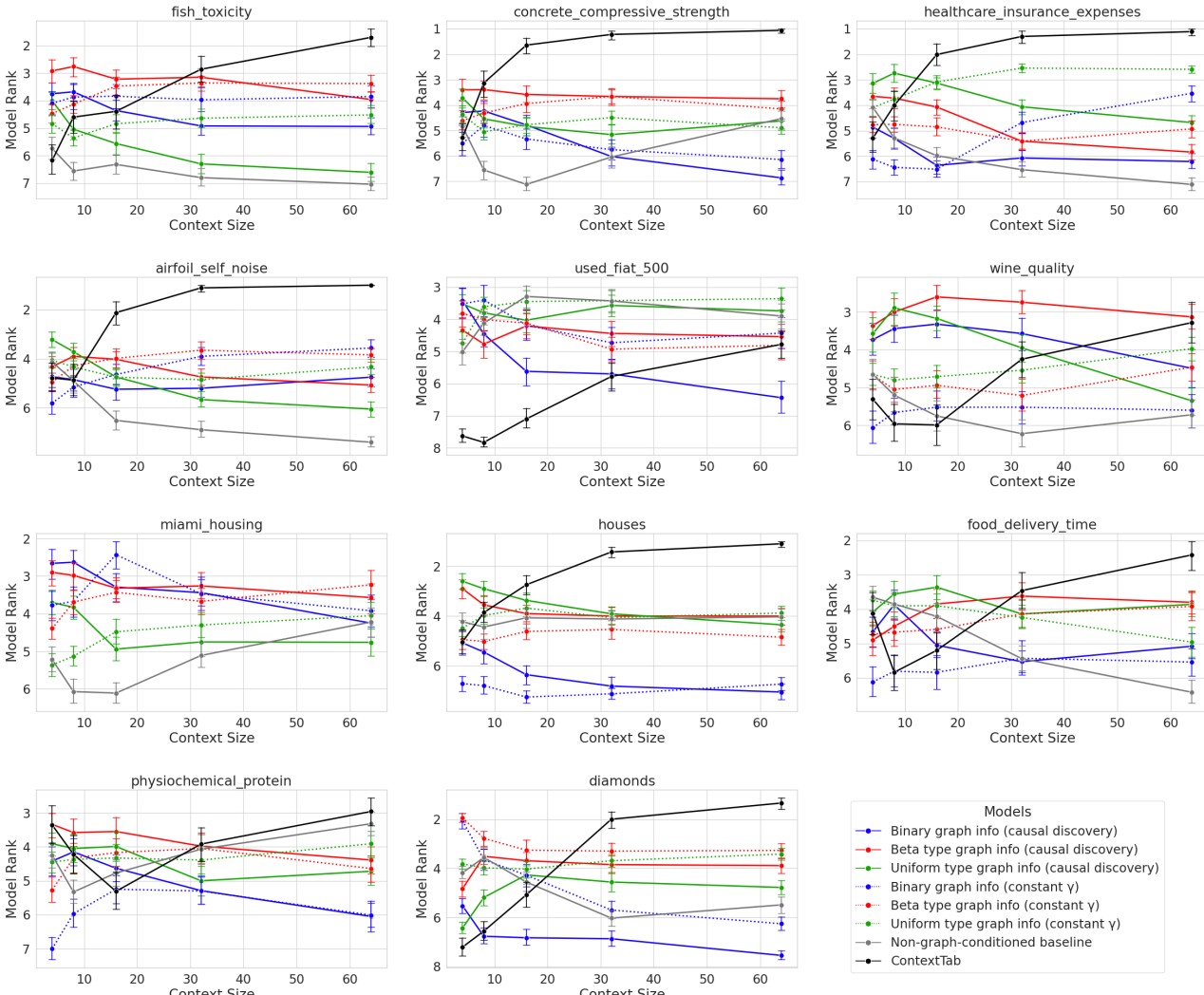

*Figure 8.* Evaluation of our models on eleven TabArena regression datasets. For each dataset, we report average model rank and bootstrapped 95% CIs over 100 subsampled datasets for each context size. As usual, each color indicates a different graph information type. The dotted lines correspond to models evaluated with the constant graph information matrix (6), the solid lines correspond to models with causal structure inferred by causal discovery. See Section E.3 for a discussion of the results.

