# OpenReview forum: "Bayesian Tabular Few-shot Learning with Causal Information"
_ICML.cc/2026/Workshop/FMSD — FMSD @ ICML 2026 Poster_

### Official Review · Reviewer_e352 · 2026-05-20

**Rating:** 7
**Confidence:** 4

**Review:**

- Summary: This paper proposes a Bayesian method for improving tabular few-shot regression by conditioning PFN-style tabular foundation models on causal graph information. The additional input is a probabilistic adjacency matrix over features and the target, representing a distribution over possible DAGs. The model is trained to approximate a graph-conditioned posterior predictive distribution, so hypotheses are weighted by both their fit to the observed context data and their compatibility with the provided causal structure. The author implement the idea using modified TabPFNv2-style architectures with either causally restricted feature-wise attention, GCN-based graph encoding with adaptive layer normalization, or both. Expriments show that causal conditioning helps most in the low-context regime and diminishes as context size increases, which matches the Bayesian intuition that the data eventually overwhelms the prior.

- Strengths: This paper is very relevant to FMSD. This work directly modifies the inference mechanism of a structured data foundation model and targets few-shot regression. The Bayesian framing is interesting and the paper makes a coherent argument: in few-shot prediction, the context set is too small to infer structural relations reliably, so supplying a distribution over causal structures should improve the posterior predictive distribution. This paper does not only assume perfect causal graphs, it studies binary, beta-type, uniform graph information, and shows that weaker probabilitic graph information can be more robust to mismatch than hard binary graph information. THis is important since real causal information is rarely exact.

- Weaknesses and Questions: As many causal papers, the main weakness is the real-world evaluation relies on fairly oracular causal information. The author infer a fixed graph using an ensemble of causal discovery methods on 500 rows per dataset, while the actual task is few-shot prediction. The author acknowledge ths limitation, but still in the real world, the users may not have either a reliable domain-expert graph or 500 extra samples for causal discovery. In addition, why does the constant uncertainty matrix sometimes outperform the non-graph-conditioned baseline on real data? This seems important and may indicate that the graph-conditioned architecture itself acts as a useful regularizer.

---

### Official Review · Reviewer_Cg2v · 2026-05-21
**Interesting application of Bayesian Tabular Prediction**

**Rating:** 9
**Confidence:** 4

**Review:**

Summary of Contributions
This paper conditions PFNs on a probabilistic distribution over causal DAGs to improve tabular few-shot regression. Three graph-conditioning architectures and three uncertainty levels are explored. The Bayesian framework formally shows the prior's influence diminishes with data size, and experiments on synthetic and real-world (TabArena) data confirm significant few-shot gains.

Strengths
- Novel and principled idea—conditioning TFMs on probabilistic causal structure is a fresh approach.
- Thorough experiments over context size, graph density, features, noise, and robustness to misspecification.
- Theory and experiments align: the prior matters most in few-shot regimes, reinforcing the Bayesian motivation.
- Real-world validation on eleven regression datasets.

Weaknesses
- Minimal weaknesses. Only comment is that the "oracular" causal information (from 500-row subsamples) may overstate practical gains.

Suggestions
- Discuss practical workflows for obtaining causal graphs in deployment.

---

### Official Review · Reviewer_MKZ5 · 2026-05-22
**Review of Bayesian Tabular Few-shot Learning with Causal Information**

**Rating:** 7
**Confidence:** 4

**Review:**

## Summary:
This paper proposes conditioning prior-data fitted networks (PFNs) on causal graph information for tabular few-shot regression. Causal structure is represented as a probabilistic adjacency matrix (γ), and the model performs Bayesian inference that weighs data-generating hypotheses by both their fit to the observed data and their compatibility with the provided causal structure. Three graph information types (binary, beta, uniform) and two architectural conditioning mechanisms (causally restricted attention, GCN-based modulation) are explored. Experiments on both synthetic and real-world data demonstrate consistent improvements over a non-graph-conditioned baseline, particularly in low-sample regimes.

## Strengths:
### 1. Theoretically grounded:
The Bayesian framing is clean and principled. Insight 2.1 about that the effect of causal conditioning diminishes as data grows is a natural and well-motivated theoretical property that is also empirically confirmed.

### 2. Real-world evaluation:
Applying the method to eleven TabArena benchmark datasets using causal discovery ensembles, rather than limiting evaluation to synthetic data, meaningfully strengthens the empirical case for the approach.

## Weakness:
### 1. Lack of other architectural exploration:
The paper modifies the TabPFNv2 architecture but does not compare against other TFMs (e.g., TabICL, TabDPT) as backbones. It is unclear whether the gains generalize across architectures or are specific to the TabPFNv2 design.

### 2. Limited discussion of failure modes:
The case of misspecified graph information is touched on briefly in the context of the three uncertainty levels, but there is no dedicated analysis in the main text of when and why providing causal information hurts performance.

## Questions for authors:
1. Would the method extend naturally to classification tasks, and if so, what would the primary obstacles be? Results on this would be interesting to add.